# Transmissible Gastroenteritis Virus: An Update Review and Perspective

**DOI:** 10.3390/v15020359

**Published:** 2023-01-27

**Authors:** Yiwu Chen, Yuanzhu Zhang, Xi Wang, Jian Zhou, Lerong Ma, Jianing Li, Lin Yang, Hongsheng Ouyang, Hongming Yuan, Daxin Pang

**Affiliations:** 1Key Laboratory of Zoonosis Research, Ministry of Education, College of Animal Sciences, Jilin University, Changchun 130062, China; 2Chongqing Research Institute, Jilin University, Chongqing 401120, China; 3Chongqing Jitang Biotechnology Research Institute Co., Ltd., Chongqing 401120, China

**Keywords:** TGEV, TGEV variant strains, TGEV virulence, alphacoronavirus, zoonotic

## Abstract

Transmissible gastroenteritis virus (TGEV) is a member of the alphacoronavirus genus, which has caused huge threats and losses to pig husbandry with a 100% mortality in infected piglets. TGEV is observed to be recombining and evolving unstoppably in recent years, with some of these recombinant strains spreading across species, which makes the detection and prevention of TGEV more complex. This paper reviews and discusses the basic biological properties of TGEV, factors affecting virulence, viral receptors, and the latest research advances in TGEV infection-induced apoptosis and autophagy to improve understanding of the current status of TGEV and related research processes. We also highlight a possible risk of TGEV being zoonotic, which could be evidenced by the detection of CCoV-HuPn-2018 in humans.

## 1. Introduction

In the past two decades, the continuous appearance of severe acute respiratory syndrome coronavirus (SARS-CoV), Middle East respiratory syndrome coronavirus (MERS-CoV), SARS-CoV-2, swine acute diarrhea syndrome coronavirus (SADS-CoV), and porcine deltacoronavirus (PDCoV) warns us that the coronavirus is still one of the biggest threats to human and livestock health [1,2,3,4,5]. TGEV is a member of the coronavirus subfamily, and although many advances in its study have been made over the past few decades, there are still many unresolved issues of viral virulence and cross-species transmission. Now it cannot be ignored that highly virulent recombinant TGEV strains are still being observed due to frequent evolution and recombination events [6,7,8,9]. Moreover, the sequencing results of the swine enteric coronavirus, SeCoV, reported in European countries, and the CCoV-HuPn-2018 virus in Malaysia both show that these newly discovered coronaviruses are inextricably linked to TGEV [10,11,12,13,14,15], and that the neglect of further work on TGEV will cause delays in public health responses to possible transboundary outbreaks.

TGEV has made entirely new advances in the wave of research on various coronaviruses [16,17,18,19]. In this review, we give a comprehensive overview of the research status of TGEV, discuss the effect of TGEV mutation on virulence, and attempt to find the patterns of change. Apoptosis is important to the process of infection in animals with the virus, may be involved in the proliferation of TGEV infection, and is also associated with severe damage to the intestinal tissue of piglets [20]. Therefore, we reviewed apoptosis induced by TGEV. In addition, we summarize the studies on the proliferation of TGEV in the host involving autophagy, in which inhibitors of the autophagy pathway can be potential antiviral drugs for TGEV. These pieces of information are hoped to help us to predict and deal with the possible future challenges in pig husbandry or new outbreaks in humans and to respond in a timely fashion to the public health problems caused by the spillover of TGEV recombinant coronavirus in the future.

## 2. TGEV Structure and Epidemiology

### 2.1. Structure of TGEV

TGEV is an alphacoronavirus with a positive-strand RNA genome of ~28.5 kb [21,22]. The entire viral genome structure includes nine open reading frames (ORF), and the arrangement is 5′ UTR-ORF1a-ORF1b-S-ORF3a-ORF3b-E-M-N-ORF7-3′UTR (Figure 1) [21,23]. ORF1a and ORF1b are involved in genome replication [22,23], occupying 2/3 of the whole TGEV genome [24]. The ORF1 function domain encodes the necessary structural protein for viral genome replication, including polymerase, helicase, and metal-binding motifs [23,25]. Additionally, ORF1 also encodes pp1a/1ab proteins after uncoating, and then 16 non-structural proteins (Nsps) are hydrolyzed by papain-like protease and 3C-like protease [26,27]. Nsp1 is one of the earliest proteins expressed by the virus in the host and is involved in multiple steps that interfere with the host immune response, inhibiting host protein production but not inducing host mRNA cleavage [27,28]. Nsp2 is the main protein involved in activating the NF-κB pathway to induce inflammation during TGEV infection [29]. Nsp3 has two papain-like protease activity domains and is involved in cleavage of Nsp2 and Nsp3 sites and may be involved in cleavage of Nsp3 and Nsp4 sites [30,31]. The main function of Nsp4 is to participate in the formation of double-membrane vesicles (DMVs) during viral infection with Nsp3 [32,33,34]. Nsp5 has protease activity and may be associated with interferon antagonism [35,36,37]. Nsp6 may also be involved in the formation of DMVs [38]. Nsp7–16 combine with each other to form a variety of multi-protein complexes involved in viral transcription and replication, and is associated with serious disruptions of intestinal homeostasis [39,40,41]. Spike (S) protein is a large transmembrane glycoprotein involved in recognizing and binding to host receptor aminopeptidase-N (APN) [42,43]. S protein has hemagglutination activity toward erythrocytes by combining sialic acid on the surface of erythrocytes [44,45]. ORF3 encodes accessory proteins ORF3a and ORF3b. Deletions in ORF3a have been reported in different TGEV and PRCV strains, which are used for homology and phylogenetic alignments between strains [7,9,46,47]. Envelope (E) protein temporarily resides in the pre-Golgi intermediate compartment during coronavirus maturation, which is a critical component for virion envelope morphogenesis [48,49]. Membrane (M) protein is related to the Golgi complex in cells. It is localized to the host Golgi when the M protein expresses alone or in the endoplasmic reticulum–Golgi intermediate compartment in TGEV-infected cells [50]. The M protein initiates virion assembly in the Golgi via virus genomic RNA and nucleoprotein, forming two distinct types of virions within the infected cells: larger virion firstly appears in the perinuclear region; smaller virion aggregates in secretory vesicles and on the cell surface [51,52]. Envelope (E) protein and M protein play an important role in the early maturation stage of the virus, which are involved in the formation of virus-like particles (VLPs) [53,54]. Therefore, the E and M proteins are indispensable in the formation of virion [55,56]. The nucleocapsid (N) protein is involved in the induction of apoptosis due to its cleavage sites of caspase-6 and caspase-7 [57,58]. ORF7 locates at the end of the viral genome and has been used for phylogenetic analysis in some research [59,60].

### 2.2. Epidemiology of TGEV and PRCV

TGEV was first reported in the United States in 1946 [61], and since then, swine transmissible gastroenteritis (TGE) has been reported successively in America, Europe, and Asia [62,63,64]. Although the prevalence of TGEV in diarrhea pigs in China is very low in recent years [65], there are still several strains prevalent in southern and northern China [7,8,9]. TGEV is divided into two classical subtypes according to the difference of the S protein and ORF3 deletion site. One is the Purdue strain with the 1123–1128nt deletion of the S protein, and this deletion feature was developed by Haelterman’s research team at Purdue University for serial passage of the virus [6,23,66,67,68]. The other is the Miller subtype with the −75–−60nt and 195–223nt deletions of ORF3 (Figure 1) [6,7,9,68].

In addition to the two classical TGEV subtypes, there is another related non-enteric pathogenic TGEV recombinant virus, porcine respiratory coronavirus (PRCV) [64,69]. PRCV is a variant of TGEV with a deletion of 621–681nt in the S protein. Unlike TGEV, PRCV is a respiratory disease (Figure 1) [64,70,71]. PRCV was first reported in Belgium in 1984 [72] and then spread to the Americas [73,74], Japan [75], the United Kingdom [76], and Hungary [77].

The lethality rate of TGEV in piglets is almost as high as 100%, and it has caused huge losses to the breeding industry in Europe, Asia, America, and other places [61,62,63,64]. The main transmission routes of TGEV are fecal–oral transmission, respiratory transmission, and breastfeeding transmission [78,79,80]. The severity of illness after TGEV infection is inversely related to the age of the infected piglets, with symptoms of vomiting, diarrhea, and dehydration two weeks after infection [79,81]. Symptoms in adult pigs include transient elevated body temperature, vomiting, diarrhea, anorexia, and sows stop lactating, with short duration and few deaths [82].

The transmission route of TGEV recombinant virus PRCV is similar to TGEV, and the main infection routes are the fecal–oral route and respiratory route through aerosol [83,84,85]. PRCV reproduces in the lungs and causes symptoms such as coughing, sneezing, fever, bronchoalveolar pneumonia, and interstitial pneumonia after infection [69,83]. Compared with the high virulence and mortality of TGEV [86,87], PRCV is less virulent and causes little pig mortality [84]. In addition, PRCV can be detected in the small intestine, but the virus does not replicate in villous epithelial cells [69,83,88].

The detection rate of TGEV has declined in the decades following the outbreak due to the continuous development of means against TGEV, such as vaccines [89,90], and the detection rate in diarrhea pigs is now very low (approximately <3%) [65,82,91]. Previous studies have shown that piglets can cross-protect against TGEV after PRCV infection, enabling piglets to effectively establish protective immune memory and survive TGEV infection [92,93]. PRCV-infected sows can also induce milk-derived immunity to TGEV in pregnancy, and immune memory can establish even in sows infected with PRCV at a young age [92,93]. Therefore, it may be exactly because of the low lethality rate of PRCV that piglets can establish immune protection against TGEV, so that TGEV does not have the same widespread epidemic effect as PEDV [84,86,92,93].

## 3. Factors Affecting TGEV Pathogenicity

### 3.1. Virulence Changes Caused by Fragment Deletion

The S protein is involved in receptor recognition and membrane fusion, delivering the nucleocapsid into host cells, which plays an important role in the virulence of the virus itself, tissue tropism in the host or cell tropism in vitro, and the pathogenesis of coronaviruses [94,95,96,97]. Reverse genetics has identified that the S1 subunit of the S protein is an important site for binding receptors. TGEV first recognizes and binds the key receptor APN on the surface of the cell through the C-terminal domain (CTD) of the S1 subunit [42,97], and then performs the process of invasion and membrane fusion to release the viral genome into the cell [97,98]. In addition, Luis Enjuanes et al. also proved that the S protein of the TGEV virus plays a crucial role in virus virulence and tissue tropism by targeted recombination of the S protein [67,99].

We could observe a classical and attenuating mutation that is considered to be typical of the Purdue strain by summarizing some of the classic virulent and attenuated strains (Table 1) [66,67]. The deletion at the 1123–1128nt site of the S protein is present in classic Purdue strains such as Purdue P115 and PUR46-MAD, and the cloned strains PTV no. C43 and C45, the 43rd and 45th generations obtained by Carlos et al. in TGEV serial passage experiments [6,23,67,68]. Purdue P115 was a clone obtained by passage 115 times on ST cells after the original TGEV strain was first isolated [100]. Thus, Purdue P115 is adapted to a specific cell by high-intensity serial passaging, resulting in the deletion of a segment of the viral genome [101,102]. However, it has been reported that not all strains carrying this 6nt deletion are all attenuated strains. For example, the SHXB strain isolated in China is considered to be a lethal virulent strain after being verified by an animal challenge test [103,104]. The specific reason for the phenomenon of strains carrying the deletion sites that were previously considered to be classical attenuated strains but still being virulent strains is not clear, which may be caused by mutations in other undiscovered sites in the viral genome. The specific mechanism of this phenomenon still needs more in-depth and specific research to verify it.

Strains carrying ORF3 −75–−60nt and 195–223nt deletion are classic Miller strains, but the deletion of these sites has not been reported to be associated with virulence. There is an attenuated strain, Miller M60, that is missing 531nt in ORF3b in the Miller strain (Table 1), and its specific deletion site is ORF3b 405–935nt, which results in a deletion of 67aa [68]. However, there is no experimental result that directly proves that the loss of 531nt in ORF3b will lead to the attenuation of virus virulence.

### 3.2. Virulence Changes Caused by Point Mutations

The factors that affect the virulence of TGEV are far more than only caused by deletion. We also describe the effect of point mutations on the S gene on virus virulence (Table 2). PRCV has been previously identified to be a variant of the 621–681nt deletion of the S protein of TGEV [64,70,71]. Based on this result, researchers focused on this deletion fragment and demonstrated through targeted recombination that changes in S gene nt655 (PTV-ts-dmar, nt G to T) and nt665 (PTV no.R29, aa L to P) [67,99] lead to attenuation of TGEV and loss of intestinal tropism.

The nt1753 base of the S protein of the firstly isolated Purdue strain is thymine, and the strains carrying T at this site, such as Purdue, TS, Miller M6, and JS2012, are all virulent strains [7]. Other strains that carry 1753T to G (aa585 S to A) mutation are attenuated strains, including attenuated TGEV strains Miller M60, Purdue P115, attenuated H, and PRCV strain PRCV-ISU-1. Therefore, the mutation of amino acid 585 from serine to alanine may be one of the reasons for the reduced virulence of TGEV strains [68]. However, some strains that also carry this mutation have been shown to be virulent, such as the AHHF and SHXB strains [6,103]. The reason for this phenomenon contradicting the previous conclusion is not clear, so whether the S585A locus can indeed lead to the attenuation of TGEV virulence needs to be proved by reverse genetics. C. Krempl et al. reported that aa145 to 155 on TGEV S protein are the sites that affect virulence, sialic acid binding sites, and hemagglutinating activity [105]. When mutations in S protein aa145 P to L, aa147 C to R, or aa155 C to F occur, TGEV loses the sialic acid binding site, hemagglutinating activity, and reduces virulence.

**Table 2 viruses-15-00359-t002:** Effect of point mutations on viral virulence and tropism.

**Strain**	**Isolate**	**Site**	**Variation**	**Changes in Virus**	**Reference**
Purdue	PTV no.R29	Spike	665nt T to C	Loss of gut tropism; Reduced virulence	[67,99]
Purdue	PTV-ts-dmar	Spike	655nt G to T	Loss of gut tropism; Reduced virulence	[67,99]
Purdue, Miller	Miller M60, Purdue P115, attenuated H, AHHF, SHXB	Spike	1753nt T to G	Reduced virulence	[7,68]
Purdue	Purdue P115	Spike	aa145 P to L; aa147 C to R; aa155 C to F	Loss of sialic acid binding site; Reduced virulence; Loss of hemagglutinating activity	[105]
Purdue	PUR46-MAD	ORF 1a	637nt G to A	Reduced virulence	[106]

The evaluation of the virulence of the virus is based on the virulence of the virus as described by the author in the first report: if the individual challenge experimental piglets die or the pathological sections of the small intestine show severe lesions, this strain is a virulent strain.

Not limited to point mutations in the S protein, Carmen et al. showed that G637A (Gly108Asp) in ORF1 of TGEV affects papain-like protease-mediated cleavage in vitro, resulting in virus attenuation [106].

## 4. TGEV Receptors

### 4.1. Aminopeptidase-N

The TGEV receptor is porcine APN (pAPN) [42]. APN is a type 2 transmembrane glycoprotein, approximately 150kDa, and it is highly expressed in intestinal, kidney, and respiratory epithelial cells. APN is mainly associated with the enzymatic regulation of peptides, viral receptors, tumor development, cell motility, and chemotaxis [107]. APN has been identified as a receptor for alphacoronaviruses from several different species, including TGEV [42,108], PRCV [108,109], human coronavirus 229E (HCoV-229E) [110], feline coronavirus (FCoV) [111,112], and canine coronavirus (CCoV) [111,113].

These alphacoronaviruses mentioned above specifically recognize the APNs of their respective host cells through triglycan proteins on the virus surface [114]. Afterwards, the S protein is divided into the S1 subunit, which specifically recognizes the cellular receptors, and the S2 subunit, which is involved in the membrane fusion, after cleavage by host cell proteases [115,116]. The viral S1 subunits have different binding sites of host APNs. Delmas et al. found that pAPN 717aa–813aa is essential for TGEV infection [117]. Kolb et al. proved that human APN (hAPN) aa288–295 is an important site for HCoV-229E by replacing the amino acid sequence QSVNETAQ (pAPN aa283–290) in pAPN with DYVEKQAS (hAPN aa288–295) [118]. Hegyi et al. found that feline APN (fAPN) aa670–840 is an important receptor binding site for FCoV [119], and Tusell et al. further narrowed the site to two regions: aa732–746 and aa764–788 [113].

Interestingly, the relationship between these alphacoronaviruses and APNs is not a one-to-one correspondence between viruses and animals. This means these viruses can bind to APNs from other species. FAPN is capable of being infected by a virus from different terminal hosts: FCoV from cats, TGEV from pigs, HCoV-229E from humans, and CCoV from dogs [111,113]. This finding suggests that when multiple alphacoronaviruses in cats are co-infected, homologous recombination may occur between similar but different strains, and recombination may produce new strains [14,112,120]. The sequencing results of disease samples collected in recent years indicate that TGEV still exists in pig husbandry and is undergoing natural recombination [7,9,10,11,12,13]. This phenomenon reminds us that the recombination and evolution of TGEV still need to be monitored. In response to the occurrence of highly pathogenic mutation events in TGEV, such as the outbreak of the highly pathogenic G2 strain of PEDV that occurred in 2010 [121,122,123] or the recombination virus that causes zoonotic diseases [14], it is necessary to maintain a high degree of vigilance in pig husbandry or public health problems.

APN is also not limited to alphacoronavirus-related receptor binding. A recent study on deltacoronaviruses showed that PDCoV can bind to pAPN through the C-terminal domain (CTD) of the S protein and use pAPN for cell entry [124]. Interestingly, Zhu et al. found that only some residues in the CTD of PDCoV S1 are involved in binding to pAPN [125]. This result is consistent with the finding that CD163 and pAPN double knockout (DKO) pigs can only reduce PDCoV susceptibility, but not block PDCoV [126]. Therefore, deltacoronaviruses can also use APN as a receptor to infect host cells like alphacoronaviruses, but cannot use APNs as a direct receptor that binds to the viral S protein and enters cells like alphacoronaviruses.

### 4.2. Non-Protein Receptors—Sialic Acid

Besides the classical TGEV receptor pAPN, the non-protein host factor sialic acid also acts as a receptor for binding to TGEV. Sialic acid is an acidic compound with a nine-carbon skeleton that can be used by the four coronaviruses of the alpha-, beta-, gamma-, and deltacoronavirus to bind and enter host cells [127,128,129,130,131,132,133,134]. TGEV is one of these coronaviruses that can use sialic acid as a non-protein receptor [44,105,135].

Unlike TGEVs, which bind to extracellular receptors using S1-CTD, TGEVs bind to sialic acid through their S1-N terminal domain (NTD) [105]. This ability to combine with sialic acid may be obtained by the ancestral coronaviruses, to use carbohydrates as receptors, steal the host galectin as a viral lectin, and then integrate it at the NTD [135,136]. The domain of sialic acid binding may be located in aa145–155 of the S protein, and point mutation or fragment deletion within this amino acid residue will lead to the loss of sialic acid binding activity (Table 2) [105]. TGEV can bind two sialic acids, N-glycolylneuraminic acid (Neu5Gc) and N-acetylneuraminic acid (Neu5Ac), and Neu5Gc is the preferred receptor for TGEV [44]. Conjugates of TGEV and sialic acid effectively protect virions in the stomach and can infect intestinal epithelial cells by binding to mucins across the mucus barrier [45]. The ability to bind sialic acid and form conjugates is dispensable for PRCVs that have lost the NT domain [105]. Although TGEV, PRCV, and HCoV-229E are all alphacoronaviruses that use APN as a receptor, PRCV and HCoV-229E are not enteropathogenic. They may have lost their intestinal tropism while losing their intact NTD. Thus, the sialic acid binding ability may be a key factor in the intestinal phase of alphacoronaviruses [44,105,137,138].

## 5. Apoptosis Associated with TGEV

TGEV invades the small intestinal epithelium after infection, causing small intestinal villous atrophy and severe diarrhea, which are central events of piglet death caused by infection. This process is closely related to apoptosis. Apoptosis is a process of programmed cell death regulated by genes, which removes damaged or virus-infected cells and plays an important role in viral infectious diseases. Apoptosis pathways are mainly divided into the mitochondrial pathway (intrinsic pathway) and the death receptor pathway (extrinsic pathway) [139,140]. We summarize the mechanism of apoptosis induced by TGEV in Figure 2.

The intrinsic apoptotic pathway is started by changes in mitochondrial membrane permeability, releasing cytochrome c (Cyt c) to activate caspase-9, and then caspase-9 activates caspase-3 to initiate the apoptotic program [141,142,143]. When caspase-3 is activated, the apoptosis program begins [144,145]. The extrinsic apoptotic pathway induces apoptosis through the specific activation of caspase-8 by death receptors, such as FasL, leading to the activation of caspase-3 [146,147]. TGEV infection promotes p53 phosphorylation and induces ROS accumulation by activated p53, leading to the translocation of Bax to mitochondria, the release of Cyt c from mitochondrial oxidative damage, and finally lead to apoptosis [141,148,149].

TGEV infection can activate not only intrinsic apoptotic pathways, but also extrinsic apoptotic pathways. After TGEV infection, caspase-8 indirectly activates caspase-9 by cleaving Bid, resulting in the release of Cyt c, or caspase-8 directly activating caspase- 3 to induce apoptosis [141]. In addition, the TGEV N protein can activate p53 and p21 during the S and G2/M phases, ultimately leading to apoptosis through the intrinsic apoptotic pathway [58]. A very interesting result related to the TGEV N protein is that caspase-6/7 has N protein cleavage activity [57], which also implies that N protein plays a crucial role in TGEV-induced apoptosis.

## 6. Autography Associated with TGEV

Macroautophagy, also known as autophagy, is a conserved degradation process in eukaryotes. Under stress conditions (such as organelle damage or pathogen invasion), autophagy-related genes (ATGs) mobilize the immune system to capture and eliminate pathogens [150,151,152]. During autophagy, the substrates to be degraded are encapsulated by autophagosomes formed by double-membrane vesicles (DMVs), then fuse with lysosomes, and finally hydrolyze by proteases in lysosomes [153,154]. Previous studies believed that autophagy, as a part of the immune system, has the function of delivering viruses into lysosomes for degradation to prevent infection, and can deliver viral antigens to the immune system [155,156,157,158]. Autophagy may have positive or negative effects on viral replication during viral infection: autophagy can be induced in TGEV infection, and the induction of autophagy can promote TGEV replication [159,160].

Several studies in recent years have demonstrated that the relationship between autophagy and viruses is not simply a positive or negative effect. With the technique called the genome-wide cellular knockout (GeCKO) library, researchers screened host factors for SARS-CoV-2 and other coronaviruses, and a brand new and poorly understood host factor in virus research, TMEM41B, came into view [16,17,18,19]. In previous studies, TMEM41B was found to be a multi-transmembrane protein localized in the endoplasmic reticulum, involved in VTT domain formation, and three independent studies have pointed out that it is involved in the early formation of autophagosomes [161,162,163]. Several recent experiments have shown that TMEM41B acts as a scramblase in the early stages of the autophagosome by flipping lipids in the lipid bilayer membrane structure [164,165], mobilizing lipids on the endoplasmic reticulum to promote membrane curvature [18,161,162,163,166], and forming the bilayer membrane of the autophagosome structure [32].

TMEM41B is critical in multiple viral infections including TGEV in multiple independent studies on multiple viruses and is an essential gene for DMV formation during coronavirus replication [16,17,18,167]. The DMV produced by coronavirus is approximately 200–400 nm in diameter, and the production of this bilayer is as closely related to the endoplasmic reticulum (ER) as the production of autophagosomes. TMEM41B does not seem to affect the binding and entry of extracellular receptors in the early stage of TGEV infection, but it mainly affects the stage of virus replication after entering cells [18,19]. This process is mainly related to the Nsp of the coronavirus. Several previous experiments have demonstrated the interaction between Nsp3 or Nsp4 and TMEM41B of several coronaviruses, including TGEV [18]. It has been proved that Nsps locate on both sides of DMV in DMV formation, which means Nsp3 and 4 are dissociated from each other [32,168]. Therefore, TGEV may take Nsp3 as the core, and through the interaction with Nsp4, hijack the ER in cells and remodel the virus adaptability, and Nsp6 may also be involved in this process [169,170,171,172,173]. After hijacking the ER, virions mobilize TMEM41B and VMP1 through the endoplasmic reticulum to recruit [32,174,175] and mobilize lipids to generate DMV structures necessary for their replication and transcription [18,19]. We summarized this progress in Figure 3. Overall, the discovery of a new host factor for TGEV, TMEM41B, expands our vision in a whole new direction. The new direction can not only promote the exploration of new mechanisms of virus infection but also allow us to have a deeper understanding of the functions of virus non-structural proteins in infection.

## 7. Conclusions

In the past 20 years, coronaviruses have emerged with alarming speed and high virulence strains, and human diseases such as SARS-CoV, MERS-CoV, and SARS-CoV2 have caused serious public health problems. TGEV is also back in our sights in the pig breeding industry and study of zoonotic disease along with the coronavirus pandemic and outbreak in recent years.

Although the detection rate of TGEV is now very low, it has been continuously detected in China [6,7,8,9], South Korea [176], and Japan [75] in recent years, and TGEV is still evolving and recombining. Naturally recombinant virulent strains JS2012 [7] and AHHF [9], with genotypes between the Purdue strain and Miller strain, were identified in southern China. Like virulent Purdue, AHHF does not carry a deletion of the Purdue signature S protein nt1123–1128 (Table 1), which may explain its high virulence. It also does not carry the deletions of Miller strains ORF3 −75–−60nt and 195–223nt (Table 1), so it was classified as a strain between Purdue and Miller clusters [9]. AHHF and SHXB carry mutations that were previously thought to cause attenuating strains, yet nevertheless show high virulence in challenge experiments. These strains were all detected and reported in China. What causes these strains to carry attenuating mutations but still show high virulence, as well as the reason why frequent recombination occurs and may produce new TGEV clusters (AHHF), needs more in-depth experiments to verify it. Not only limited to the recombination evolution of the TGEV virus itself, Italy [10], Germany [13], Spain [11], and Denmark [12] have also reported the swine enteric coronavirus (SeCoV) composed of natural recombinant strains of PEDV and TGEV. The S gene of SeCoV has more than 90% base identity with PEDV, and the rest of the viral genome has up to 97% base identity with TGEV virulent strains H16 and Miller M6 [10]. These important issues in pig farming caused by TGEV cannot be ignored. The evolution and mutation of these strains may lead to the impact of vaccines and other effective control methods.

TGEV is a cross-species infectious disease; we need to increase our focus on TGEV in this time of accelerated occurrence of zoonotic diseases. The canine–feline-like recombinant alphacoronavirus was recently reported for the first time in a case of pneumonia in Malaysia and was named CCoV-HuPn-2018 [14]. This virus has a CCoV S1 domain and a FCoV S2 domain, and sequence alignment analysis shows that it also has more than 90% similarity with the whole genome of TGEV Purdue (virulent). It is very interesting, and not to be overlooked, that TGEV, CCoV, and FCoV can all use fAPN to enter host cells [111,113]. This study not only reports the emergence of what may be an eighth human disease-causing coronavirus, but that it is also a very dangerous signal of possible future public health problems. In the past, we have rarely conducted in-depth research on the potential threats related to cats and dogs and their corresponding coronaviruses, and the research on pigs and their related coronaviruses has been limited to just solving the problem of economic losses caused by coronaviruses to the breeding industry. We did not pay attention to the hidden dangers that these coronaviruses may cause. The emergence of SeCoV and CCoV-HuPn-2018 reminds us that even though TGEV exists in nature with extremely low activity now, it is still undergoing continuous recombination and evolution. We should not only concentrate on the economic damage that TGEV has caused, but also on the public health problems it may cause in the future as a transboundary infectious disease.

In the recent future, exploring the host factors of TGEV that have not been covered based on GeCKO library technology provides insight into the infection mechanism of this classic alphacoronavirus. It will be important to conduct deeper information mining based on the results and discover and identify these gene functions, which have hardly been studied in the virus field before. Meanwhile, the research results that have been reported may continue to provide new clues for the biology of coronaviruses or other naturally recombined novel coronaviruses in the future. Moreover, these clues allow us to revisit and gain a deeper understanding of this virus that has been quietly lurking in nature for some time, which has implications for the possible future (and possibly contemporary) emergence of a zoonotic recombinant alphacoronavirus and its impact on human public health.

## Figures and Tables

**Figure 1 viruses-15-00359-f001:**
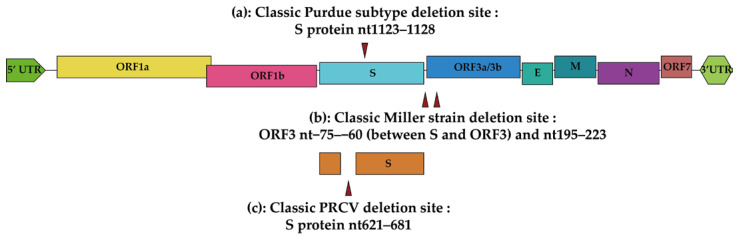
Schematic diagram of the TGEV genome structure. Non-structural proteins, including ORF1a and 1b, ORF3a/3b and ORF7; structural proteins, including spike (S), envelope (E), membrane (M), and nucleocapsid (N) proteins. S protein is used to divide different TGEV subtypes: (**a**) nt1123–1128 deletion of the S protein is considered to be Purdue strain; (**b**) ORF3 nt−75–−60 and nt195–223 deletions (between S protein and ORF 3) are considered to be Miller subtypes; (**c**) and TGEV with a deletion of nt621–681 in S protein is variant strain PRCV and infects the porcine respiratory tract.

**Figure 2 viruses-15-00359-f002:**
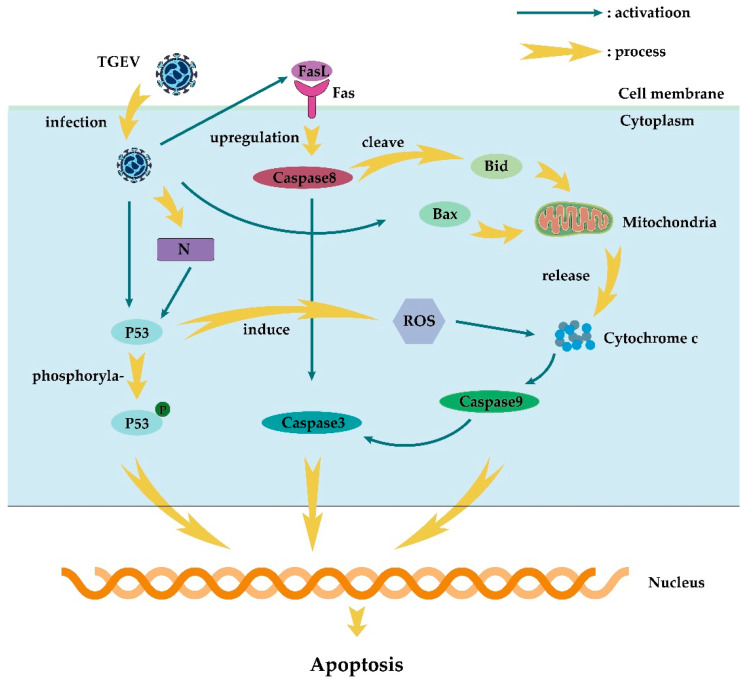
Diagram of the roles of apoptosis in the TGEV infection.

**Figure 3 viruses-15-00359-f003:**
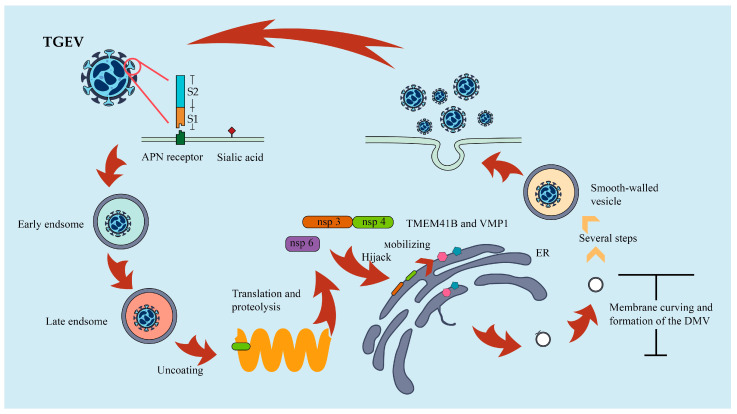
Schematic diagram of the role of autophagy in TGEV in infected host cells. TGEV infects host cell through the S protein recognition receptor pAPN, and this process can also be achieved by the non-protein receptor sialic acid. Then, it is entered and released through the endosomal pathway. Nsp3 and Nsp4 are hydrolyzed and bound to the endoplasmic reticulum, hijacking the host endoplasmic reticulum and recruiting TMEM41B and VMP1 to form the membrane vesicles necessary for viral replication, a process that may involve Nsp6.

**Table 1 viruses-15-00359-t001:** Representative strains, classic variation, and pathogenicity of each TGEV genotype.

**Strain**	**Isolate**	**Variation**	**Pathogenicity**	**Genbank ID**	**Reference**
Purdue	Virulent Purdue	NA	Virulent	DQ811789.2	[6,68]
Purdue	Purdue P115	S protein nt1123–1128 deletion	Attenuated	DQ811788.1	[6,68]
Purdue	PUR46-MAD	S protein nt1123–1128 deletion	Attenuated	NC_038861.1	[6,23]
Purdue	SHXB	S protein nt1123–1128 deletion	Virulent	KP202848.1	[103,104]
Between Purdue and Miller	AHHF	S protein nt2386–2388 deletion	Virulent	KX499468.1	[6,9]
Miller	H16	ORF3 −75–−60nt and 195–223nt deletion, S protein nt2386–2388 deletion	Virulent	FJ755618.2	[6,9]
Miller	Miller M60	ORF3 −75–−60nt and 195–223nt deletion, ORF3b 405–935nt deletion	Attenuated	DQ811786.2	[6,68]
Miller	Miller M6	ORF3 −75–−60nt and 195–223nt deletion	Virulent	DQ811785.1	[6,68]
Miller	JS2012	ORF3 −75–−60nt and 195–223nt deletion	Virulent	KT696544.1	[6,7]
Miller	attenuated H	ORF3 −75–−60nt and 195–223nt deletion, S nt2386–2388 deletion	Attenuated	EU074218.2	[9]

The evaluation of the virulence of the virus is based on the virulence of the virus as described by the author in the first report: if the individual challenge experimental piglets die or the pathological sections of the small intestine show severe lesions, this strain is a virulent strain.

## Data Availability

Not applicable.

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
