# Peer review of "Transmissible Gastroenteritis Virus: An Update Review and Perspective"

_viruses, 2023, doi:10.3390/v15020359_

Round 1

Reviewer 1 Report

This manuscript describes about genetic variants and their pathogenicity, and host receptors and mechanism of apoptosis of TGEV infection. The properties of TGEV are briefly summarized in this text, but their descriptions are not enough. Therefore, this manuscript needs to improve according to reviewer's comments.

Figure 1

·        Classical Purpue  Classical Purdue

·        Please change the color of ORF7, because difficulty of read.

Page 4, line 141-142

What are PTV no.C43 and C45?

Please reconsider this sentence.

Table 1

·        Pathogenicity

Please add the criteria of the pathogenicity of strains listed in this Table 1 in the legend.

·        Two TGEV strains (H16 and AHHF) have different deletions compared with other Purdue strains, but these strains belong into Purdue strains. Please describe the reason in detail.

·        Four Miller strains include in Table 1, but the characteristics of Miller strains have not in the text. Please add the explanation.

Table 2 and 3.2 Virulence changes caused by point mutations

The effects of point mutations in S gene into virulence are described in the text, which the strains presented in the text are not matched those listed in Table 2.

Please improve the difference.

Page 5, line 181

Please correct misspelling.

5.     Apoptosis

Why did the authors select the topic of TGEV “apoptosis”?

Please explain the reason in Introduction.

6. Autography

The authors describe association of NSP proteins with the role of autography.

However, the origin and function of NSP protein dose not describe in the text.

Therefore, the authors should origin and function of NSP proteins in detail.

Author Response

Dear editor and reviewers:

Thank you very much for your comments concerning our manuscript entitled “Transmissible gastroenteritis virus: an update review and perspective” (viruses-2117871). Those comments are all valuable and very helpful for revising and improving our paper, as well as the important guiding significance to our researches. We have revised the manuscript accordingly (with blue fonts in the text) and a detailed response to the reviewers’ comments has been provided below.

Responses to Reviewer’s Comments: “Transmissible gastroenteritis virus: an update review and perspective” (viruses-2117871)

To Reviewer #1:

  1. * Figure 1
  • Classical Purpue → Classical Purdue
  • Please change the color of ORF7, because difficulty of read.

Response: Thanks very much for your good advice and careful review. We sincerely apologize for my spelling mistakes in fig 1 and for choosing colors in fig 1 that make the text difficult to read. In the revised manuscript, we have revised and adjusted it in fig 1.

  1. * Page 4, line 141-142

What are PTV no.C43 and C45? Please reconsider this sentence.

Response: Thanks very much for your good advice. PTV no. C43 and C45 are the 43rd and 45th generation strains obtained by the researchers in the continuous passage of TGEV in the experiment, which belong to the strains obtained by human intervention, and do not belong to the wild-type strains. After your reminder, we are working for PTV no. The sources of C43 and C45 are supplemented in more detail in the revised manuscript.

  1. * Pathogenicity

Please add the criteria of the pathogenicity of strains listed in this Table 1 in the legend.

Response: Thanks very much for your good advice. In the revised manuscript, we evaluate the virulence of the virus is based on the virulence of the virus as described by the author in the first report: If the individual challenge experimental piglets die or the pathological sections of the small intestine show severe lesions, this strain is a virulent strain.

  1. * Two TGEV strains (H16 and AHHF) have different deletions compared with other Purdue strains, but these strains belong into Purdue strains. Please describe the reason in detail.

Response: Thanks very much for your good advice and careful review. In the revised manuscript, we first classified the H16 strain into the Miller strain because it carries ORF3-75-60nt and 195–223nt deletions, which are typical Miller strain characteristics. Once again, we apologize for the misinformation in the previous form. Second, the reason we previously classified AHHF as a Purdue strain is because it does not carry ORF3-75-60nt and 195–223nt deletions and is not a miller strain. After considering the question you raised, we concluded that it should be classified as between Purdue and Miller clusters based on the results of the phylogenetic tree constructed by the original article. Because AHHF is not in a group with other Purdue strains used for phylogenetic tree construction, nor with other Miller strains in the phylogenetic tree. (Zhang et al. Identification of a natural recombinant transmissible gastroenteritis virus between Purdue and Miller clusters in China. Emerg Microbes Infect 2017, 6, e74, doi:10.1038/emi.2017.62.)

  1. * Four Miller strains include in Table 1, but the characteristics of Miller strains have not in the text. Please add the explanation.

Response: Thanks very much for your good advice. In the revised manuscript, we supplemented the description of Miller strain characteristics after describing the characteristics of the Purdue strain and supplemented a deletion site specifically carried by Miller M60.

  1. * Table 2 and 3.2 Virulence changes caused by point mutations

The effects of point mutations in S gene into virulence are described in the text, which the strains presented in the text are not matched those listed in Table 2.

Please improve the difference.

Response: Thanks very much for your good advice. We have revised and adjusted the content of the table in response to your comments. In the revised manuscript, the content in table 2 better matches the strains mentioned in the text, and we have supplemented the information in table 2 and the main text to complete the Purdue P115 content that was previously inadvertently omitted.

  1. * Page 5, line 181

Please correct misspelling.

Response: Thank you very much for your careful review. Did you refer to the misspelling "cell motility"? This word refers to the movement of cells. For this usage, we refer to the title and content of the article "Aminopeptidase N is involved in cell motility and angiogenesis: Its clinical significance in human colon cancer" before writing it in the review. If it's not this error, we hope you can point us to the typo in this line and we will correct it immediately.

  1. * Apoptosis

Why did the authors select the topic of TGEV “apoptosis”?

Please explain the reason in Introduction.

Response: Thanks very much for your good advice. Apoptosis is important to the process of infection in animals with the virus, may be involved in the proliferation of TGEV infection, and is also associated with severe damage to the intestinal tissue of piglets. Therefore, we chose this topic for review. In the revised manuscript, we have explained the reason in Intoduction.

  1. * Autography

The authors describe association of NSP proteins with the role of autography.

However, the origin and function of NSP protein dose not describe in the text.

Therefore, the authors should origin and function of NSP proteins in detail.

Response: Thanks very much for your good advice. After discussion, we believe that the lack of information about the virus NSP does cause a barrier to reading, thank you again for your valuable comments. In the revised manuscript, we supplement and discuss the sources and functions of 16 nsps of TGEV based on previous studies, and these are placed in "2.1. Structure of TGEV".

Reviewer 2 Report

This paper reviews the basic biological properties of TGEV, factors affecting virulence, viral receptors, and the lastest research advances in TGEV infection-induced apoptosis and autophagy. Which provides a comprehensive understanding of TGEV biology, natural recombination and the mechanisms of induction of apoptosis and autophagy in hosts

However, the review does not provide a comprehensive and in-depth overview of the latest research progress of TGEV, such as virus infection mechanism, cross-species infection, and virus recombination, and lacks more detailed progress of research results. Moreover, The TGEV structure and epidemiology, pathogenicity, viral receptors, etc. mentioned in the manuscript are the basic information known about TGEV, which has been reported in the published literratures.  (Kim et al., 2000; Hu et al., 2015; Liu et al., 2021). Therefore, the review lacks an overview of the latest research advances in TGEV and is not recommended for publication in current manuscript in the Viruses journal. this manuscript may be reconsidered after major revisions. 

Author Response

Dear editor and reviewers:

Thank you very much for your comments concerning our manuscript entitled “Transmissible gastroenteritis virus: an update review and perspective” (viruses-2117871). Those comments are all valuable and very helpful for revising and improving our paper, as well as the important guiding significance to our researches. We have revised the manuscript accordingly (with blue fonts in the text) and a detailed response to the reviewers’ comments has been provided below.

Responses to Reviewer’s Comments: “Transmissible gastroenteritis virus: an update review and perspective” (viruses-2117871)

To Reviewer #2:

Response: Thanks very much for your good advice and careful review. We carefully read the "Kim et al., 2000; Hu et al., 2015; Liu et al., 2021" three articles. Kim et al., 2000's paper compared the molecular characterization and pathogenesis of TGEV and PRCV, and focus on prior PRCV or concurrent TGEV/PRCV infections in a herd may influence and change the pathogenesis of TGEV. Hu et al., 2015's paper focused on the TGEV-HX strain isolated in China. The full paper is mainly based on the whole genome sequencing data of TGEV-HX to explore the development, evolution and recombination information of this strain. The main purpose of this study is to analyze the genetic variation of TGEV in China provides essential information for further understanding the evolution of TGEVs. And Liu et al., 2021's paper mainly reviewed the pathogenicity, prevalence and diagnostic methods of TGEV, SADS-CoV, PDCoV, and PEDV, but does not focus on TGEV this one PEC.

In our review, we focus on the recombination of TGEVs, factors influencing TGEV virulence, and recent advances in TGEV-related research. On the issue of viral recombination and cross-species infection, we discussed the possibility of cats as TGEV hosts and highlights the potential for TGEV recombination during cross-species transmission in "4.1. Aminopeptidase-N", which could pose a potential risk to future animal husbandry or human public health problems. In addition, we discussed in "7. Conclusion" the CCoV-HuPn-2018 coronavirus discovered in Malaysia in 2018, which infects humans and has a very close relationship with TGEV, FCoV, CCoV, highlighting the need for a more positive attitude towards TGEV and not just focus on the problems it poses in the livestock industry. This also re-emphasizes the problem of cross-species infection of TGEV. Based on the results of host factor screening of TGEV by genome-wide cellular knockout (GeCKO) library technology in recent years, combined with the latest progress of autophagy in coronaviruses, we also discussed and summarized the relationship between host factor TMEM41B and the non-structural proteins nsp3,4,6 of TGEV. It’s an important advance in the mechanism of viral infection hosts discovered more than seventy years since TGEVs were first isolated. Because the virus infection process involved in TMEM41B is not only applicable to the TGEV virus, but also plays a very meaningful guiding role in the prevention and control of other coronaviruses.

Our review focuses on TGEV as a virus, which helps to provide readers with more detailed and in-depth information about TGEV, rather than just scratching the surface. This information will guide researchers on possible future outbreaks of public health problems caused by TGEVs.

Round 2

Reviewer 1 Report

Please  correct the misspelling "Purpue" into "Purdue" in Figure 1.

Reviewer 2 Report

The revised manuscript has been sufficiently improved to publication in Viruses.